# Mesenchymal Stem Cells Influence Activation of Hepatic Stellate Cells, and Constitute a Promising Therapy for Liver Fibrosis

**DOI:** 10.3390/biomedicines9111598

**Published:** 2021-11-02

**Authors:** Chanbin Lee, Minju Kim, Jinsol Han, Myunghee Yoon, Youngmi Jung

**Affiliations:** 1Department of Integrated Biological Science, Pusan National University, Pusan 46241, Korea; lcb102@pusan.ac.kr (C.L.); kmj0802@pusan.ac.kr (M.K.); wlsthf1408@pusan.ac.kr (J.H.); 2Division of Hepatobiliary and Pancreas Surgery, Department of Surgery, Biomedical Research Institute, Pusan National University, Pusan 46241, Korea; ymh@pusan.ac.kr; 3Departments of Biological Sciences, Pusan National University, Pusan 46241, Korea

**Keywords:** hepatic stellate cells, liver fibrosis, mesenchymal stem cells, cell-free therapy, extracellular vesicles

## Abstract

Liver fibrosis is a common feature of chronic liver disease. Activated hepatic stellate cells (HSCs) are the main drivers of extracellular matrix accumulation in liver fibrosis. Hence, a strategy for regulating HSC activation is crucial in treating liver fibrosis. Mesenchymal stem cells (MSCs) are multipotent stem cells derived from various post-natal organs. Therapeutic approaches involving MSCs have been studied extensively in various diseases, including liver disease. MSCs modulate hepatic inflammation and fibrosis and/or differentiate into hepatocytes by interacting directly with immune cells, HSCs, and hepatocytes and secreting modulators, thereby contributing to reduced liver fibrosis. Cell-free therapy including MSC-released secretomes and extracellular vesicles has elicited extensive attention because they could overcome MSC transplantation limitations. Herein, we provide basic information on hepatic fibrogenesis and the therapeutic potential of MSCs. We also review findings presenting the effects of MSC itself and MSC-based cell-free treatments in liver fibrosis, focusing on HSC activation. Growing evidence supports the anti-fibrotic function of either MSC itself or MSC modulators, although the mechanism underpinning their effects on liver fibrosis has not been established. Further studies are required to investigate the detailed mechanism explaining their functions to expand MSC therapies using the cell itself and cell-free treatments for liver fibrosis.

## 1. Introduction

Chronic liver disease caused by hepatitis C virus infection, alcohol abuse, metabolic disease, or nonalcoholic hepatitis is a global health threat [1,2,3,4]. In fact, the number of patients suffering from liver disease is growing rapidly, and approximately 2 million people die from it worldwide every year [5,6,7]. Liver fibrosis is a result of chronic damage and inflammation, characterized by the excessive accumulation of extracellular matrix (ECM) [8,9,10]. Persistent damage replaces functional hepatic cells with ECM proteins that distort the liver structure and functions, leading to cirrhosis or liver cancer [11,12,13,14]. Hepatic stellate cells (HSCs) are the major contributors to liver fibrosis [15]. In a damaged liver, HSCs transdifferentiate into activated HSCs that produce ECM proteins, such as collagen and fibronectin [16,17]. Hence, regulating HSC activation has been considered as a potential strategy to prevent the progression of liver disease [18]. However, an effective treatment for liver fibrosis has not yet been established to date, although several therapies to modulate HSC activity have been attempted for the treatment of liver fibrosis.

Among various efforts to alleviate liver fibrosis, stem cell therapy is considered to be a promising therapeutic approach. Mesenchymal stem cells (MSCs) are post-natal stem cells found in almost all tissues and have therapeutic potential for the treatment of liver fibrosis [19]. They possess self-renewal capacity, differentiate into multiple cell types, and secrete anti-apoptotic and immunomodulatory molecules [20,21,22,23]. Given these abilities, many clinical trials using MSCs have been conducted for various diseases, such as neurological disorders, diabetes, cardiac infarction, and liver disease [24,25,26,27]. Several experimental and clinical studies have supported that MSC-mediated therapy has potential benefits for treating liver fibrosis/cirrhosis. For instance, Jang et al. [28] reported that the treatment of bone marrow (BM)-derived MSCs alleviated alcohol-related hepatic fibrosis in humans. Yao et al. [29] showed that transplantation of human placental MSC relieved carbon tetrachloride (CCl_4_)-induced liver fibrosis and HSC activation by increasing caveolae 1. Notably, in addition to treatments using the MSCs themselves, MSC-released factors have proven their therapeutic potential for liver diseases. Herein, we summarized the pathogenesis of liver fibrosis focusing on HSCs and reviewed MSC applications for the treatment of liver fibrosis to elucidate their therapeutic potential.

## 2. Pathogenesis of Liver Fibrosis and Activation of HSCs

Liver fibrosis is a wound-healing process that occurs in response to liver damage [15]. However, continuous and/or severe damage would impair the hepatic function and architecture, leading to liver fibrosis defined as excessive ECM accumulation [9,10,11]. The liver is mostly composed of hepatocytes that perform a wide range of hepatic functions, such as detoxification, bile acid synthesis, and various metabolism processes [11,30]. These are the first cells exposed to most types of hepatic injuries [31]. In response to mild injury, hepatocytes proliferate and replace the damaged tissues. However, persistent and/or severe injury would exceed the regeneration capacity of the hepatocytes and give rise to a massive hepatocyte death [15,32]. Instead of hepatocytes, non-parenchymal cells proliferate and fill the parenchymal areas where the hepatocytes were originally situated with excessive ECM proteins, leading to liver fibrosis. Hence, liver fibrosis impairs the hepatic function and architecture, leading to death from liver failure (Figure 1).

HSCs are the main producers of ECM proteins in the liver [15]. In a normal liver, HSCs are located at the subendothelial space between the hepatocytes and the sinusoidal endothelial cells in a quiescent state [33]. Quiescent HSCs are non-proliferative and are characterized by storing retinoic esters and expressing glial fibrillary acidic protein [34,35]. With the occurrence of liver damage, HSCs gradually lose these distinctive features. They undergo transdifferentiation into myofibroblast-like HSCs in a process called activation [16,36]. Activated HSCs have higher capacities for contractility, proliferation, and migration and exhibit a concurrent immense alternation of the gene expression profile. These cells are able to synthesize ECM proteins such as collagen type I (Col I), fibronectin, and α-smooth muscle actin (α-SMA) [17]. HSC activation is triggered first by paracrine stimulation from neighboring cells in the damaged liver. Injured hepatocytes release various cytokines/chemokines and reactive oxygen species, which recruit immune cells into the injured lesion and stimulate HSC activation [15,37,38]. Recruited immune cells and Kupffer cells, liver-resident macrophages produce inflammatory cytokines such as interleukin (IL)-1β and IL-6 [8]. These cytokines are important paracrine signals to promote HSC activation. Activated HSCs maintain their activated status in an autocrine manner by secreting profibrotic factors and enhancing hepatic fibrosis [39]. Numerous pro-fibrotic factors, such as transforming growth factor-β (TGF-β), Hedgehog (Hh), and platelet-derived growth factor (PDGF), are known to promote the transcription of fibrotic genes in HSCs [40,41,42]. TGF-β plays a crucial role in controlling HSC activation and ECM accumulation in the liver [43,44]. TGF-β induces the phosphorylation of SMAD, the downstream signaling molecule of TGF-β [40]. The phosphorylated SMAD translocates into the nucleus, where it promotes the transcription of fibrotic genes, such as collagen, α-SMA, and tissue inhibitor of metalloproteinases (TIMPs), during HSC activation. In addition, TGF-β enhances autophagy influx in HSCs through ERK and JNK pathways, and leads to HSC activation [45]. Hh ligands released from dying hepatocytes stimulate the activation and/or proliferation of Hh-responsive cells, such as immune cells, HSCs, and progenitors [46,47,48,49]. Upon binding of the Hh ligand to Patched, a receptor of Hh ligands, another receptor called Smoothened (Smo) is activated to induce the translocation of a Gli-krüppel family member (Gli) into the nucleus, acting as a transcriptional factor for Hh signaling and profibrotic genes [41,50]. In a fibrotic microenvironment, hepatocytes that escaped the TGF-β killing become responsive to Hh and undergo epithelial-mesenchymal transition (EMT) [51,52,53,54]. During EMT, Hh-responsive hepatocytes lose their adhesion capacity and acquire an ECM-producing myofibroblast phenotype, functioning as a source of EMT production. In addition, Hh signaling recruits inflammatory cells and upregulates their production of pro-inflammatory cytokines, such as IL-6, tumor necrosis factor (TNF)-α, and TGF-β, eventually contributing to the accumulation of fibrous ECM in the liver [53,55,56].

## 3. Basic Information of MSCs Focusing on Therapeutic Potential

MSCs are one of the adult stem cells that can be isolated from almost all tissues, including the BM, adipose tissue, placenta, umbilical cord, and other tissues [57]. They were first introduced from BM by Friedenstein and his colleagues in 1968 [58]. These cells proliferate in vitro as plastic-adherent heterogenous cells with spindle-shaped fibroblast-like morphology [59]. MSCs express the antigen cluster of differentiation (CD)105, CD73, and CD90, but they lack CD45, CD34, and human leukocyte antigen (HLA) class II. MSCs are multipotent cells that have the potential for self-renewal and the ability to differentiate into mesoderm lineages, including adipocytes, chondroblasts, and osteoblasts [21,60]. Furthermore, MSCs are capable of differentiating into several somatic cells, such as neural cells, myoblasts, and hepatocytes [61,62,63]. Unlike embryonic stem cells (ESCs) and induced pluripotent stem cells (iPSCs), MSCs are easily isolated from various tissues and safely harvested without ethical issues [64]. In addition, MSCs can avoid T cell recognition and immune responses due to their low expression of major histocompatibility complex (MHC) class II and immunosuppressive effects [23,65]. These unique immunotolerant phenotypes of MSCs allow the possibility of using allograft for patients. Hence, MSC administration has emerged as a promising therapeutic candidate for stem cell-based therapy because of its powerful advantages for clinical use as regenerative medicine. 

Growing evidence demonstrates that MSC administration has therapeutic benefits for liver disease. In 70% partially hepatectomized rats, MSC was shown to enhance liver regenerative capacities by facilitating glucose and lipid metabolism [66,67]. The transplantation of human adipose-derived MSC reduced hepatic ischemia-reperfusion injury by improving cell proliferation in rats [68]. Umbilical cord-derived MSC suppressed lipid accumulation in obese type 2 diabetic mice by promoting β-oxidation and suppressing lipogenesis [69]. Meanwhile, tonsil-derived MSCs alleviated liver fibrosis through the exosomes [70]. In this section, we reviewed the therapeutic effects of MSCs themselves and MSC-derived factors on reducing liver fibrosis, specifically focusing on HSC activation.

## 4. MSC Administration for Liver Fibrosis

The transdifferentiation of quiescent HSCs into myofibroblastic HSCs is a major event leading to liver fibrosis. Hence, regulating HSC activation has been considered as the therapeutic target of liver fibrosis. Recent findings show that MSCs inhibit HSC activation by either suppressing proliferation or stimulating apoptosis, exerting an anti-fibrotic action (Figure 2). In a culture system that allowed MSCs to be in contact directly with HSCs, the MSCs suppressed HSC proliferation by upregulating the Notch 1 expression and downregulating the PI3K/Akt pathway, a critical pathway inducing HSC proliferation in HSCs, although the exact mechanism by which Notch1 decreased p-Akt in HSCs has not been elucidated [71]. It has also been reported that MSCs prevented HSCs from entering the S phase by upregulating the inhibitors of cell proliferation, such as p27^Kip1^ and p21^Cip1^, and downregulating the accelerators of cell cycle, namely, cyclin D and p-ERK [72]. The phosphorylated ERK1/2 that is highly associated with HSC proliferation was also found to be reduced in HSCs co-cultured with MSCs. In addition, the apoptosis of activated HSCs is promoted, and their viability is significantly alleviated through direct or indirect contact with MSCs. When activated HSCs were co-cultured with human BM-MSCs, the HSCs exhibited a significant increase of pro-apoptotic proteins, including Bax and cleaved caspase-3, compared with the single cultured cells [73]. Lin et al. [74] demonstrated that increased apoptosis of HSCs after being co-cultured with MSCs was mediated by the nerve growth factor (NGF) signaling. In NGF signaling, the NGF secreted from MSCs inhibited nuclear factor kappa B (NF-κB) and reduced the expression of anti-apoptosis gene *B cell leukemia-xl* (*Bcl-xl*). MSCs also release the hepatocyte growth factor (HGF), which inhibited the NF-κB pathway and attenuate fibrogenic characteristics in LX2, a well-established human HSC line [75]. Consistent with in vitro results, the transplantation of MSCs ameliorated the formation of liver fibrosis in CCl_4_-injected rats by inhibiting the proliferation and promoting the apoptosis of HSCs [76]. 

MSCs indirectly suppress HSC activation by interacting with immune cells or changing themselves (Figure 2). To reduce profibrotic stimulation from immune cells toward HSCs, MSCs alleviate immune responses by using their immunosuppressive feature and/or differentiation ability into hepatocytes to replace damaged hepatocytes. The immunosuppressive property of MSCs is well documented, and treatments using MSC immunomodulation have been widely studied in various diseases, including diabetes, Alzheimer’s, inflammatory bowel disease, and osteoporosis [77,78,79,80]. In liver fibrosis, MSCs have been shown to alleviate the expression of inflammatory factors inducing HSC activation [81]. MSCs also exert anti-inflammatory properties by releasing anti-inflammatory cytokines IL-4 and IL-10, promoting immune cells to secret anti-inflammatory cytokines [82,83,84,85]. Furthermore, they inhibit the synthesis of pro-inflammatory cytokines such as TNF-α, interferon (IFN)-γ, and IL-17 in various types of immune cells, including T cells, natural killer (NK) cells, neutrophils, and Kupffer cells [85]. In CCl_4_-induced liver fibrosis mice, intravenously injected mouse BM-MSCs successfully migrated to the damaged liver and significantly relieved liver fibrosis by decreasing IL-17 and increasing IL-10 [86]. The immunomodulatory capacity of MSCs is further evidenced by the production of major effector molecules prostaglandin E2 (PGE2) and indoleamine 2,3-dioxygenase (IDO). The increased release of PGE2 by MSCs significantly suppresses the proliferation and/or activation of T cells, monocytes, NK cells, and macrophages including Kupffer cells [87,88,89,90]. IDO is known to be an important suppressor of effector T cells by metabolizing tryptophan, which is essential for T cell effector function [86]. In an MSC-transplanted liver with fibrotic injury, the serum level of IDO increased, the amount of IL-17, a key cytokine for T cell activation and neutrophil mobilization, decreased, and proliferation of T helper 17 (Th17) cells was suppressed [91]. Eventually, immune response was alleviated. In addition, MSCs themselves contribute to immunosuppression by escaping from immunity because of low MHC class molecule expression [23,65].

Given that hepatic fibrosis is initially induced by the death of hepatocytes, the replacement of damaged hepatocytes can be a fundamental therapy for hepatic fibrosis. Since MSCs can differentiate into diverse types of cells, the differentiation of MSCs into hepatocyte-like cells is a promising strategy to replace damaged or dying hepatocytes. A treatment involving a combination of factors including HGF, fibroblast growth factor (FGF) 2/4, and dexamethasone induces MSCs to differentiate into hepatocyte-like cells [63,92,93]. In in vitro systems, differentiation induces MSCs to express hepatocyte markers, such as albumin and α-fetoprotein, and show hepatocytic functions including glycogen synthesis, albumin secretion, low-density lipoprotein (LDL) uptake, urea production, and activation of cytochrome P450 activity which is critical for drug metabolism [63,94,95]. Several groups have also demonstrated the hepatic differentiation of MSCs in experimental animal models and humans. Transplantation of MSC-derived hepatocyte-like cells attenuated liver fibrosis and improved liver function in CCl4-injured mice [96]. All of these findings clearly prove that MSCs exert therapeutic effects on HSCs directly or indirectly through cell-to-cell contact and paracrine signaling. MSC-based experiments have been summarized in Table 1.

MSC engineering that modifies gene expression or metabolic process impacting biological functions of MSCs could increase proliferative and immunomodulatory properties of MSCs, enhancing their therapeutic function. Among factors engineering MSCs, TGF-β was shown to remarkably increase MSC proliferation [97,98,99,100]. TGF-β1 treatment elevated BM-MSCs proliferation by promoting nuclear localization of β-catenin in Smad3-dependent manner [99]. Kim et al. [100] reported that TGF-β1 stimulated expression of runt-related transcription factor 1, and extended self-renewal and proliferation of MSCs. MSCs engineered with TGF-β also reinforced their therapeutic efficacy for several diseases, such as type 1 diabetes, sepsis and renal ischemia/reperfusion injury, by increasing their immunomodulatory potential, although they rarely had effect on liver fibrosis. TGF-β-manipulated MSCs increased the insulin production and inhibited the expressions of pro-inflammatory cytokines in mice with type 1 diabetes [101]. Liu et al. [102] revealed that MSCs overexpressing TGF-β have more favorable therapeutic effects by decreasing macrophage infiltration in cecal ligation and puncture-induced sepsis mice than MSCs. Administration of TGF-β-overexpressed MSCs restored renal function and attenuated inflammation against renal ischemia/reperfusion injury [103]. In addition, autophagy is known to be a cell protection mechanism against various harmful damages, suggesting that autophagy influx promotes MSC viability, and contributes to the enhanced therapeutic potential of MSCs. Regmi et al. [104] demonstrate that elevated autophagy increases cell viability and decreases ROS generation of three-dimensional cultured MSCs (MSC_3D_), and the MSC_3D_ alleviates dextran sulfate sodium-induced colitis damage more effectively than two-dimensional cultured MSCs (MSC_2D_) do. It was also shown that MSC_3D_ had more resistance to severe oxidative stress than MSC_2D_ did, and that MSC_3D_ effectively modulated inflammation and improved therapeutic effect in the acute liver failure model [105]. In addition, autophagy regulates immunomodulatory effect of MSCs. Gao et al. [106] have revealed that rapamycin (autophagy inducer) increased immunosuppressive potential of MSCs, whereas 3-MA (autophagy inhibitor) reduced it. These findings suggest that enhanced autophagy influx in MSCs increase the therapeutic potential of MSCs. However, Wang et al. [107] presented that suppression of Beclin-1, essential mediator of autophagy, in MSC improved therapeutic and immunomodulatory properties of MSCs in CCl_4_-induced liver fibrosis model. The results are contrary to other findings. Therefore, further study is required to determine the effect of autophagy in MSC efficacy.

Although the transplantation of MSCs is considered a safe and efficient application for chronic liver fibrosis, there are still several limitations. Cell-based therapy, including stem cell therapy, satisfies complex but appropriate regulations, such as safety, purity, and potency, for human applications [108,109]. Furthermore, MSC transplantation requires a large number of cells, reaching as many as hundreds of millions, and takes about 10 weeks to reach the needed number of MSC before transplantation [110,111,112]. Furthermore, the efficacy of MSC treatment is still questionable due to low engraftment, abnormal differentiation, and risk of tumor growth after transplantation [113,114,115,116]. The MSC-mediated inhibition of HSC activation has been mainly proven in relatively constant in vitro systems. However, the anti-fibrotic effect of MSCs differs depending on the employed animal models and the experimental methods of MSC transplantation. In light of the foregoing, further studies are required to overcome the obstacles in applying MSC-based therapies for the treatment of liver fibrosis. 

## 5. MSCs-Based Clinical Application for Liver Disease

Fifty-six clinical trials using MSCs for liver disease, such as cirrhosis, acute liver failure, and hepatitis, have been reported and the majority of these were registered in Asian countries [117]. MSCs that are mostly obtained from bone marrow, umbilical cord, and adipose tissue are used in clinical trials of the liver, and injected through the peripheral vein or the hepatic artery [118]. The number of cells and frequency of injection varies, and the MSC effect on liver fibrosis is inconsistent. Body-weight-based dosing within the range (0.5 × 10^6^–3 × 10^6^ cells/kg) for a single dose is used in most clinical trials, where some studies use total MSC quantity (1 × 10^7^–20 × 10^7^ cells). Doses as low as 1 × 10^7^ MSCs showed the significant attenuation of liver fibrosis in some cases [119], while a higher dose of 2 × 10^8^ MSCs rarely improved liver fibrosis in others [120]. However, more than 2 × 10^8^ MSCs reduced liver damage in patients with liver cirrhosis [121]. In most clinical studies, one-time doses were administrated, but had no significant difference in treating liver cirrhosis compared with two doses a month apart [122]. Nevertheless, most clinical trials have shown that MSC-based therapies have beneficial effects on liver fibrosis (Table 2). In patients with alcoholic liver cirrhosis, hepatic arterial injection of BM-MSCs significantly improved liver function and reduced collagen accumulation [28,123]. Patients with HBV-induced liver cirrhosis exhibited alleviated expression of fibrotic markers and remarkable decrease of model of end-stage liver disease (MELD) score at 48 weeks after MSC injection, although the level of HBV DNA in serum was not changed [124,125]. It was also reported that MSC treatment attenuated liver cirrhosis by exerting their immunomodulatory properties. MSC transplantation upregulated immunomodulatory factors, such as serum M-colony stimulating factor, macrophage migration inhibitory factor, and IL-18, and regulated imbalance of Treg/Th17 cell [119,126]. In addition, the protective effect of hepatocyte-like cells differentiated from MSCs was reported in several clinical trials. Hepatocyte-lineage committed MSCs presented similar protective effects like undifferentiated MSCs [127], and they improved liver function in patients with liver cirrhosis or end-stage of liver disease [127,128,129]. Furthermore, no side effects or complications were observed in the above clinical studies except for mild fever that occurred after MSC injection. The mild fever disappeared within 12 h after MSC treatment. However, two papers have reported that MSCs rarely have therapeutic effect in patients with liver cirrhosis. Mohamadnejad et al. [120] presented that MSC injection hardly reduced liver damage compared with the placebo group at 48 weeks after injection. Kantarcioglu et al. [130] also showed that scores of MELD and Child-Pugh in cirrhosis patients did not change before and after MSC treatment. Therefore, it is necessary to conduct further study for large-scaled clinical trials encompassing multiple conditions, such as a wider range of doses and frequencies, and various administration routes, to establish the effective therapeutic dose and frequency for the clinically safe and long-term effect of MSC in liver disease.

## 6. MSC Cell-Free Therapy for Liver Fibrosis

MSCs have been shown to reduce liver fibrosis and promote liver regeneration despite their low engraftment, and their therapeutic potential is based on the paracrine effect. Conditioned media (CM) from MSCs are known to regulate HSC activation and reduce liver fibrosis. MSC-CM inhibit HSC activation by reducing pro-fibrotic gene expressions, such as *α-SMA*, *Col I*, and *matrix metalloproteinase*
*(MMP) 2*, in TGF-β-treated human primary HSCs [131]. This in vitro finding has also been proven in in vivo models by presenting reduced collagen accumulation and inflammation, as well as elevated hepatocyte survival in the livers of CCl_4_-injected mice treated with MSC-CM [132]. These data clearly indicate that MSC-CM contains many beneficial substances, suggesting that their use can be an alternative approach for MSC transplantation because the problems caused by using MSCs themselves can be avoided. MSC-free therapies are more economical and safer for clinical applications. 

MSCs secrete a variety of factors, such as cytokines/chemokines, free nucleic acids, extracellular vesicles (EVs), and lipids in response to physiological or pathological stimuli [133,134]. These MSC-derived secretomes and EVs have similar therapeutic functions as MSC-based therapies (Figure 3, Table 3). The tumor necrosis factor-inducible gene 6 protein (TSG-6) is one of the anti-inflammatory cytokines secreted by MSCs [135]. Recently, it has been demonstrated that TSG-6 decreases HSC activation and promotes the transdifferentiation of activated human primary HSCs into functional stem-like cells, thereby alleviating liver fibrosis [136]. The treatment of TSG-6 has also been shown to induce M2 polarization and MMP12 expression in macrophages [137]. Increased MMP12 suppresses HSC activation by restraining the release of pro-inflammatory cytokines. Notably, the anti-fibrotic effect of TSG-6 has been confirmed by other groups presenting that TSG-6-depleted MSCs lost their anti-fibrotic action in fibrotic livers of mice. The milk fat globule EGF factor 8 protein, a cytokine released from MSCs, downregulated the expression of the TGF-β receptor TGFβR1 by binding αvβ3 integrin to the HSCs, thus protecting against hepatic fibrosis [138].

EVs are membrane-bound vesicles that include apoptotic bodies (50–4000 nm), microvesicles (100–1000 nm), and exosomes (40–100 nm) [139]. EVs are proven to have similar beneficial functions as their parental MSCs and play a critical role in cell–cell communication [140,141]. Growing evidence shows that EVs derived from MSCs have a therapeutic effect in liver fibrosis. Exosomes derived from MSCs significantly reduced hepatocyte death and oxidative stress in a CCl_4_-induced liver fibrosis model [142]. Moreover, MSC-exosomes have been shown to alleviate liver fibrosis by inactivating the TGF-β/SMAD signaling pathway in CCl_4_-damaged liver [143]. In addition, Rong et al. [144] reported that MSC-derived exosomes inhibited HSC activation and improved liver function by suppressing the Wnt/β-catenin pathway in both activated HSCs and fibrotic tissues, and even their effects were significantly better than using MSC itself. In addition, EVs exerted a protective effect by transferring their various beneficial cargoes, such as microRNA (miR) and soluble proteins, to the target cells and tissues. Anti-oxidative glutathione peroxidase 1 delivered by the MSC-exosomes decreased oxidative stress and increased hepatocyte proliferation in CCl_4_-injured liver [145]. Furthermore, the treatment of MSCs-EV containing miR-150-5p reduced the expression of CXC chemokine-ligand-1, one of the profibrotic chemokines in HSCs, and attenuated liver fibrosis [146]. The miR-1246 contained in MSC-exosomes was shown to protect hepatocytes from ischemia-reperfusion injury and modulate the balance of regulatory T cells and Th17 cells to suppress inflammation and maintain immune tolerance [147]. Our group also reported that MSC-derived EVs had a high level of miR-486-5p, and the delivery of miR-486-5p to the fibrotic livers of mice attenuated HSC activation and liver fibrosis by abrogating Hh signaling [148]. The miR-125b in MSC-exosomes was reported to target Smo and inactivate HSCs by blocking Hh signaling [149].

Although MSC-EVs retain various cargoes with biologically beneficial effects, MSC-derived EVs are at the early stages of applications in clinical trials for various diseases. However, there are currently no clinical trials for liver disease. Many major obstacles, such as their application strategies and treatment efficacies, as well as the stable and consistent obtainment of EVs, remain to be resolved [150,151]. MSC-derived EVs are highly dynamic because EV production and their bioactive cargos in EVs are closely related to the biological microenvironment of the parent cells, namely, the MSCs [152,153]. This implies that EV production and its contents change depending on the biological status of the MSCs. The current technologies for EV isolation, including chromatography, ultrafiltration, centrifugation, and chemical precipitation, have very low yields and require a long time and a large-scale MSC culture system to obtain a defined dose of EVs for investigating therapeutic effects [112,154]. Therefore, the standardization of an experimental approach securing an optimal quality and quantity of EVs to ensure a consistent therapeutic effect of MSC-derived EVs is necessary. In in vivo models, EVs are mainly delivered to the liver, intestine, lung, and spleen [155]. It is worth mentioning that EV distribution depends on the dose and injection route. Therefore, EV delivery efficiency needs to be enhanced so that they can be delivered to the potential target cells/tissues. In addition, due to the lack of studies on the precise molecular mechanism through which the injected EVs are accepted by the target cells, organ- or cell-specific molecular signatures that recognize EVs should be elucidated.
biomedicines-09-01598-t003_Table 3Table 3Summary of MSC cell-free therapy for liver fibrosis.TypeTreatmentTarget/Experimental ModelMechanismOutcomeRefIn vitroMSC-CMHSCsReducing expressions of pro-fibrotic genesInhibition of HSCs activation[131]Cytokine, TSG-6HSCsDecreasing HSC activation Promoting the transdifferentiation of activated HSCs into stem-like cellsAmelioration of liver fibrosis[136]MSC-exosomesHSCsInhibiting Wnt/β-catenin signalingInhibition of HSC activation[144]Co-culture with miR-125b manipulated MSCsLX2Targeting Smo;suppression of Hh signalingInhibition of HSC activation[149]In vivoMSC-CMCCl_4_-induced liver fibrosis in miceReducing collagen accumulation and inflammationElevating hepatocyte survivalAmelioration of liver fibrosis[132]Cytokine, TSG-6CCl_4_-induced liver fibrosis in miceInducing M2 polarizationUpregulating MMP12 expression in macrophagesAmelioration of liver fibrosis[137]Cytokine, MFGE 8TAA or CCl_4_-induced liver fibrosis in miceDownregulating the expression of TGFβR1 of HSCAmelioration of liver fibrosis[138]MSC-exosomesCCl_4_-induced liver fibrosis in miceReducing hepatocyte deathDecreasing oxidative stressAmelioration of liver fibrosis[142]CCl_4_-induced liver fibrosis in miceInactivating TGF-β/SMAD signaling pathwayAmelioration of liver fibrosis[143]CCl_4_-induced liver fibrosis in ratsSuppressing Wnt/β-catenin signalingImprovement of liver function[144]GPX1 delivered by the MSC-exosomesCCl_4_-induced liver fibrosis in miceDecreasing oxidative stress Increasing hepatocyte proliferationAmelioration of liver fibrosis[145]miR-150-5p contained in MSC-exosomesCCl_4_-induced liver fibrosis in miceReducing the expression of CXCL-1Amelioration of liver fibrosis[14]miR-1246 contained in MSC-exosomesischemia/reperfusion-induced liver injury in miceModulating the balance of regulatory T cells and Th17 cellsProtection of hepatocytes Suppression of inflammation[147]miR-486-5p contained in MSC-exosomesCCl_4_-induced liver fibrosis in miceSuppressing Hh signalingAmelioration of liver fibrosis[148,149]miR-125b contained in MSC-exosomesCCl_4_-induced liver fibrosis in ratsMSC, Mesenchymal stem cell; HSC, Hepatic stellate cell; CM, Conditioned media; TSG-6, Tumor necrosis factor-inducible gene 6 protein; miR, MicroRNA; Hh, Hedgehog; Smo, Smoothened; MMP12, Matrix metalloproteinase 12; MFGE8, Milk fat globule EGF factor 8 protein; CCl_4_, Carbon tetrachloride; TAA, Thioacetamide; TGFβR1, Transforming growth factor beta-receptor 1; TGF-β, Transforming growth factor beta; GPX1, Glutathione peroxidase 1; Th17, T helper 17; CXCL-1, CXC chemokine-ligand-1.


To overcome the multiple challenges for the therapeutic application of MSC-EVs, researchers have attempted to improve the yield and effectiveness of EVs. For example, to increase the amount of EVs obtained, tangential flow filtration (TFF), which induces EVs to pass through membranes to filter specific EVs, has emerged as a powerful and scalable technique [156,157]. TFF reportedly produces higher-yield and higher-activity EVs than ultracentrifugation does [158,159]. To boost the therapeutic potential of MSC-EVs, engineering MSC-EVs have been manufactured. These EVs have enhanced stability and more beneficial cargo. MiR-122-loaded MSC-EVs inhibit the proliferation of HSCs by suppressing the target genes, such as insulin-like growth factor receptor 1 and Cyclin G1 [160]. In addition, miR-181-5p-modified EVs derived from MSCs have been reported to inhibit HSC activation by targeting Bcl-2 and STAT3 [161]. Manipulated EVs loading a higher level of Insulin growth factor like-1 showed remarkable anti-fibrotic effects by inactivating the HSCs and reducing the production of pro-inflammatory cytokines in macrophages compared with native EVs [162]. In addition, EVs covered with polyethylene glycol hydrogel were shown to have increased bioavailability and anti-fibrotic effects compared with native EVs [163]. The membrane modification of EVs also seems to be applied to enhance their delivery efficacy. Yang et al. [164] demonstrated that membrane-edited EVs harboring the virus fusogen increased cargo transfer into the target cells by interacting with LDL receptors. Modified EVs with cationized pullulan showed enhanced accumulation in the liver, especially in the hepatocytes, rather than in other organs, thereby reducing liver inflammation [165]. 

## 7. Conclusions

Liver fibrosis is one of the public health burdens for which effective drugs for reversing/eliminating ECM accumulation are unavailable [7]. In particular, considering that liver fibrosis is accompanied in most liver diseases, the development of a therapeutic agent for liver fibrosis is urgent and essential. Until recently, liver transplantation has been the most effective way to treat end-stage liver fibrosis. However, it could cause side effects, such as higher recipient demand compared with the number of liver donors, infection, and immune rejection [166,167]. Hence, studies exploring effective strategies that regulate the activation of HSCs are warranted to lay the foundation for the development of therapeutics against liver fibrosis. 

MSCs have emerged as an attractive application with therapeutic potential for the treatment of liver disease, including liver fibrosis. Accumulating evidence demonstrates that MSC-based therapy is a clinically relevant solution based on its interesting properties, including its abilities of differentiation and immunomodulation and the availability and ease of harvesting. However, there are obstacles in clinical applications due to the limitations of MSCs. Further studies are required to overcome these limitations by finding the most functional and accessible sources of cells, determining the optimal transplantation conditions, and increasing the regenerative abilities of transplanted MSCs in the damaged tissues. MSC-released secretomes and EVs have emerged as acellular regenerative medicines that go beyond the limitations of MSC-based therapy. However, current studies have not fully deciphered the biological active molecules and the mechanisms underlying their anti-fibrotic effects. Before adopting them as a clinical approach, further investigations should be conducted to understand the characteristics, therapeutic potential, and quantification of MSC-secretomes and EVs. In conclusion, all of the successful achievements in this field to date indicate the possibility of MSCs constituting an effective therapeutic agent for liver fibrosis. The development of safer and highly effective strategies emphasizing the anti-fibrotic effects of MSCs is required.

## Figures and Tables

**Figure 1 biomedicines-09-01598-f001:**
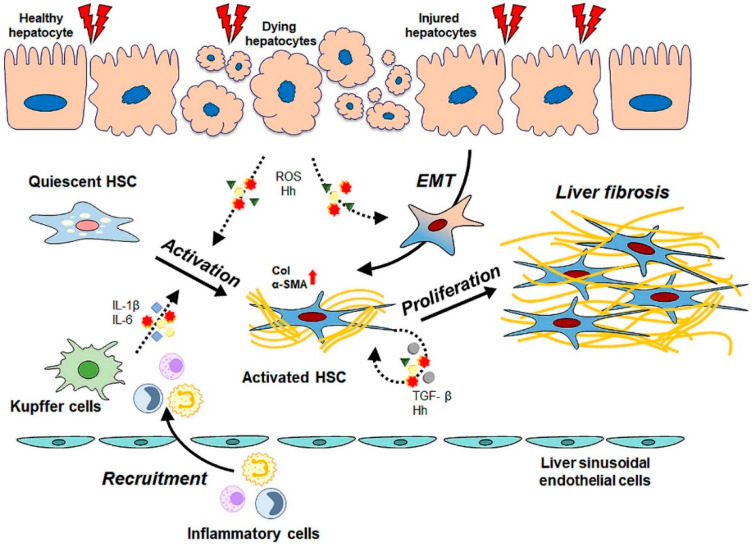
Cellular and molecular pathophysiology of liver fibrosis. In healthy liver, hepatic stellate cells (HSCs) that reside in the subendothelial space are non-proliferative and quiescent. When the liver is damaged, apoptotic hepatocytes release various cytokines and reactive oxygen species (ROS), which induce recruitment of inflammatory cells and HSC activation. Activated HSCs and have proliferative phenotype and act as a main producers of extracellular matrix (ECM) proteins such as collagen (Col), and α-smooth muscle actin (α-SMA). The recruited inflammatory cells produce pro-inflammatory cytokines such as interleukin (IL)-1β and IL-6 to trigger activation of HSCs. HSC activation is initiated by stimulation by paracrine signals secreted by surrounding cells. Activated HSCs maintain an activated phenotype in response to pro-fibrotic factors such as transforming growth factor-β (TGF-β) and hedgehog (Hh) in an autocrine manner. These profibrotic factors also induce morphological alterations in hepatocytes through the epithelial-mesenchymal transition (EMT), accelerating the accumulation of ECM proteins. Eventually, excessive deposition of ECM proteins produced by activated HSCs impairs hepatic functions and structure, leading to liver fibrosis and cirrhosis.

**Figure 2 biomedicines-09-01598-f002:**
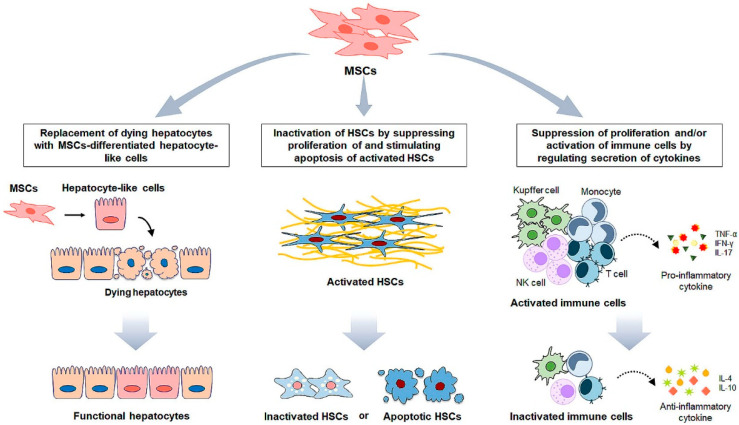
MSC-based therapeutic approaches in liver fibrosis. Administration of mesenchymal stem cells (MSCs) exerts therapeutic effects for the liver fibrosis through various mechanisms. MSCs can replace dying hepatocytes by differentiate into hepatocyte-like cells. MSC-derived hepatocyte-like cells express hepatocyte markers, and show hepatocytic functions including glycogen synthesis, albumin secretion, low-density lipoprotein (LDL) uptake, urea production and drug metabolism. MSCs inhibit HSC activation by suppressing proliferation of and stimulating apoptosis of activated HSCs. MSCs upregulate the inhibitors of cell proliferation and downregulates the accelerators of cell cycle. Increased proapoptotic proteins by MSCs also reduce viability of activated HSCs viability. In addition, MSCs alleviate immune responses by interacting immune cells. MSCs exert anti-inflammatory properties by suppressing the synthesis of pro-inflammatory cytokines such as tumor necrosis factor (TNF)-α, interferon (IFN)-γ, and IL-17, and promoting production of anti-inflammatory cytokines, IL-4 and IL-10, in immune cells such as T cell, natural killer (NK) cells, neutrophils and Kupffer cells. In addition, MSCs significantly reduce proliferation and/or activation of immune cells to suppress immune response.

**Figure 3 biomedicines-09-01598-f003:**
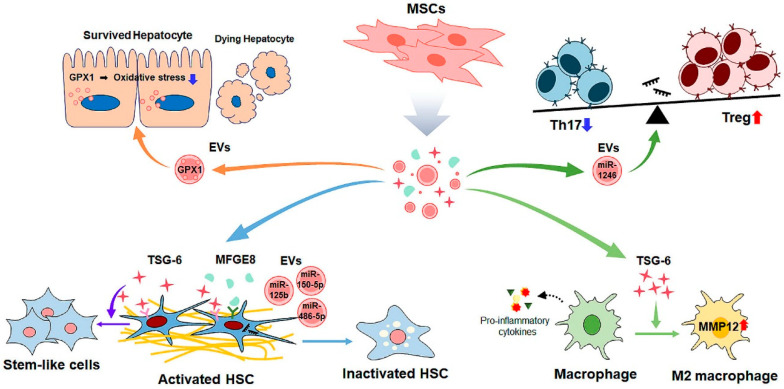
Therapeutic strategy of MSC-derived secretome and EVs for liver fibrosis. MSCs-derived secretome and extracellular vesicles (EVs) have therapeutic potential for liver fibrosis. In hepatocyte, MSCs-EV delivering glutathione peroxidase 1 (GPX1) reduces oxidative stress, and inhibits hepatocyte death, attenuating liver fibrosis. Several cytokines and EVs targeting HSC also ameliorate liver fibrosis. Tumor necrosis factor-inducible gene 6 protein (TSG-6) induces trans-differentiation of activated HSCs into stem-like cell. Milk Fat Globule EGF Factor 8 protein (MFGE8) and MSCs-EV carrying microRNA (miR) such as miR-486-5p, miR-150-5p and miR-125b reduce liver fibrosis through inactivating HSCs. In addition, MSC-EVs regulate the proportion of T cell by decreasing T helper 17 (Th17) cells and increasing regulatory T (Treg) cells. In macrophage, TSG-6 triggers polarization of M2 macrophage and upregulates MMP12 which suppresses HSC activation.

**Table 1 biomedicines-09-01598-t001:** Summary of MSC administration for liver fibrosis.

Type	Treatment	Target/Experimental Model	Mechanism	Outcome	Ref
In vitro	Direct contact with MSCs	HSCs	Upregulating Notch1 expression Downregulating PI3K/Akt pathway	Suppression of HSC proliferation	[71]
	Co-culture with MSCs	HSCs	Upregulating p27^kip1^ and p21^cip1^ Downregulating cyclin D and p-ERK		[72]
	Co-culture with MSCs	HSCs	Increasing pro-apoptotic proteins (Bax and cleaved caspase-3)	Increase of apoptosis of activated HSCs	[73]
	Co-culture with MSCs	HSCs	Secreting NGF; inhibition of NF-κB Decrease of *Bcl-xl* expression		[74]
	Direct contact or co-culture with MSCs	LX2	Releasing HGF; inhibition of NF-κB		[75]
	Co-culture with MSCs	KCs	Promoting secretion of anti-inflammatory cytokines	Alleviation of immune response	[82]
	Co-culture with MSCs	NK cells	Producing PGE2; impairment of proliferation and activation of NK cells		[89]
	Hepatocyte differentiating factors	Hepatocytes	Inducing expression of hepatocyte markers (albumin, α-fetoprotein)	Differentiation of MSCs into functional hepatocytes	[63,92,93,94,95]
In vivo	MSC transplantation	CCl4-induced liver fibrosis in rats	Inhibiting proliferation and promoting apoptosis of activated HSCs	Amelioration of liver fibrosis	[76]
	MSC transplantation	DMN-induced liver fibrosis in rats	Releasing IL-4 and IL-10	Alleviation of immune response and liver fibrosis	[82]
	MSC transplantation	CCl4-induced liver fibrosis in mice	Decreasing IL-17 Increasing IL-10		[86]
	MSC transplantation	CCl4-induced liver fibrosis in mice	Increasing IDO level and decreasing IL-17; decrease of proliferation of Th17 cells		[91]
	MSCs derived hepatocyte-like cells transplantation	CCl4-induced liver fibrosis in mice	Mimicking hepatocyte functions	Amelioration of liver fibrosis Improvement of liver function	[96]

MSC, Mesenchymal stem cell; HSC, Hepatic stellate cell; HGF, Hepatocyte growth factor; NF-κB, Nuclear factor kappa B; Bcl-xl, B cell leukemia-xl; KC, Kupffer cell; IL, Interleukin; Th17, T helper 17; NK, Natural killer; CCl_4_, Carbon tetrachloride; DMN, Dimethylnitrosamine; PGE2, prostaglandin E2; IDO, Indoleamine 2,3-dioxygenase.

**Table 2 biomedicines-09-01598-t002:** Summary of clinical trials using MSCs for liver disease.

Clinical Trials					
	Patients	Dose /Frequency	Administration Route	Outcome	Ref
Therapeutic effects	MSC injection				
	11 patients with alcoholic cirrhosis	5 × 10^7^ MSCs Two times	Hepatic artery	Decrease of MELD and Child-Pugh score Downregulation of collagen accumulation	[28]
	55 patients with alcoholic cirrhosis	5 × 10^7^ MSCs One or two times	Hepatic artery	Decrease of Child-Pugh score Decrease of ALP Downregulation of collagen accumulation	[123]
	45 patients with HBV liver cirrhosis	0.5 × 10^6^ MSCs/kg Three times	Peripheral vein	Increase of serum albumin Decrease of total bilirubin Decrease of MELD Na score Downregulation of serum laminin	[124]
	43 patients with HBV-induced acute-on-chronic liver failure	0.5 × 10^6^ MSCs/kg Three times	Peripheral vein	Increase of serum albumin, cholinesterase Decrease of total bilirubin and ALT Increase of survival rate Decrease of MELD score	[125]
	39 patients with HBV liver cirrhosis	Unknown	Hepatic artery	Increase of serum albumin Decrease of total bilirubin Decrease of MELD score Ameliorating imbalance of Treg/Th17 cells	[119]
	4 patients with liver cirrhosis	3.3 or 6.6 × 10^5^ MSCs/kg One time	Hepatic artery	Increase of serum albumin Elevating immunomodulatory factors	[126]
	Administration of hepatocyte-like differentiated MSCs				
	25 patients with HCV liver cirrhosis	1 × 10^6^ cells/kg One time	Peripheral vein	Increase of serum albumin Decrease of serum creatinine, total bilirubin Decrease of MELD score	[127]
	8 patients with end-stage of liver disease	3–5 × 10^7^ cells One time	Peripheral vein	Decrease of serum creatinine Decrease of MELD score	[128]
	40 patients with HCV-induced end-stage liver disease	2 × 10^7^ cells One time	Intrasplenic or intrahepatic	Increase of serum albumin Decrease MELD and Child-Pugh score	[129]
No effect	27 patients with liver cirrhosis	1.2–2.95 × 10^8^ MSCs One time	Peripheral vein	No changes in serum albumin, ALT and MELD scores	[120]
	25 patients with liver cirrhosis	1 × 10^6^ MSCs/kg One time	Peripheral vein	No change in serum ALT, ALP, total bilirubin MELD, and Child-Pugh scores	[130]

MSC, mesenchymal stem cell; MELD, model for end-stage liver disease; ALP, alkaline phosphatase; HCV, hepatitis C virus; HBV, hepatitis B virus; ALT, alanine aminotransferase; Treg, regulatory T cells; Th17, T helper 17; IL, interleukin.

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
