# Peer review of "Mesenchymal Stem Cells Influence Activation of Hepatic Stellate Cells, and Constitute a Promising Therapy for Liver Fibrosis"

_biomedicines, 2021, doi:10.3390/biomedicines9111598_

Round 1

Reviewer 1 Report

This is a timely review on an important issue "Mesenchymal stem cells influence activation of hepatic stellate cells". They have focused on fibrosis and involvement of stellate cells. I have an important suggestion that needs to be applied. 

We all know about the impact of TGF-beta on liver fibrosis. Also we all know TGF-beta not only induces fibrosis, but also autophagy. I have strong  recommendation to respected authors to include a section how cross talk of TGF-beta/autophagy is potentially involved in regulation of MSC induced stellate cells activation and fibrosis. This section should include scheme for better understanding for the readers. 

Author Response

Several studies have reported that the increased autophagy flux in HSCs provides energy for HSCs activation [Redox Biol. 2017 Apr; 11():322-334./ Mol Biol Rep. 2021; 48(2): 1915–1924./ Gastroenterology. 2012 Apr;142(4):938-46]. Given that enhanced autophagy promotes HSC activation and liver fibrosis, autophagy inhibition in activated HSCs is suggested as one of therapeutic strategies against liver fibrosis. [J Hepatol. 2011; 55: 1353-1360/ Biol Pharm Bull. 2014;37(9):1505-9/Gastroenterology. 2012 Apr; 142(4):938-46]. Liver fibrosis is a complex process in which diverse growth factors and cytokines are involved with a complex interplay between diverse hepatic cells and various signaling pathways, such as TGFβ, Hedgehog, PDGF, VEGF, and so on [Nat Rev Gastroenterol Hepatol. 2017 Jul;14(7):397-411.]. As you pointed out, TGF-β is a very well-known pro-fibrotic factor to orchestrate liver fibrogenesis. TGF-β regulates HSCs activation and ECM accumulation in the liver by upregulating fibrotic markers. Recently, it is shown that TGF-β increases autophagy flux during HSCs activation and promotes HSCs activation through ERK and JNK pathway [Int J Mol Med. 2021 Jan;47(1):256-266]. Considering that TGF-β-induced autophagy promotes HCS activation and MSCs have anti-fibrotic effect by targeting TGF-β, the possibility of the inhibitory effect of MSCs on TGF-β-induced autophagy in HSCs is predicted. However, there is still no report that MSCs alleviate HSC activation by suppressing TGF-β-induced autophagy in HSCs. Thus, we could not explain how TGF- β /autophagy is potentially involved in regulation of MSC induced HSCs and liver fibrosis, although further studies are needed to reveal the therapeutic role of MSC by targeting TGF-β-induced autophagy during HSCs activation. Therefore, we explained briefly that TGF-β promotes HSC activation by enhancing autophagy in HSCs in the revised manuscript; “In addition, TGF-β enhanced autophagy influx in HSCs through ERK and JNK pathways, and led to HSC activation.”

Reviewer 2 Report

In the manuscript entitled “Mesenchymal stem cells (MSCs) influence activation of hepatic stellate cells, and constitute a promising therapy for liver fibrosis” the authors report the findings presenting the activated hepatic stellate cells based pathogenesis of liver fibrosis and therapeutic effects of MSCs, MSCs-derived secretome and Cell free extracellular vesicles (EVs) strategies for the treatment of liver fibrosis. Overall, this paper will be useful to the readers of this journal, however, minor comments need attention:.

  1. It would be better if the paper reports in tabular form and compare different strategies e.g. MSCs, MSCs-derived secretome and EVs (separately as in vitro, in vivo and clinical studies), source of MSCs, their potential outcome, and underlying mechanisms, pros and cons of each strategy in tabular form.
  2. In addition to pre-clinical studies, it should include clinical trials studies (type of clinical trial, no. of patients treated/control, route, dose, patient response etc) using MSCs based and cell free therapeutic agents for treatment of liver fibrosis.

https://hrjournal.net/article/view/4205

https://inflammregen.biomedcentral.com/articles/10.1186/s41232-017-0045-6

https://www.hindawi.com/journals/sci/2021/6546780/

https://www.ncbi.nlm.nih.gov/pmc/articles/PMC7339207/

https://www.e-cmh.org/journal/view.php?number=1579

  1. The report may include information as to which source of MSCs or MSCs derived exosomes is best for the treatment of liver fibrosis/cirrhosis and why?

Author Response

  1. It would be better if the paper reports in tabular form and compare different strategies e.g. MSCs, MSCs-derived secretome and EVs (separately as in vitro, in vivo and clinical studies), source of MSCs, their potential outcome, and underlying mechanisms, pros and cons of each strategy in tabular form.

: Thank you for your helpful comments. In the revised manuscript, we added tables (Tables 1 and 3) summarizing the anti-fibrotic effect of MSCs. We categorized their effects on the liver fibrosis by types of studies, experimental conditions, mechanisms and results. If we added all information you requested in the table, it is too bustling and complex. Hence, we organized the important information in the tables so that they are clearly visible at a glance. In addition, we added the information of clinical studies in the table 2 and section 5.

  1. In addition to pre-clinical studies, it should include clinical trials studies (type of clinical trial, no. of patients treated/control, route, dose, patient response etc) using MSCs based and cell free therapeutic agents for treatment of liver fibrosis.

https://hrjournal.net/article/view/4205 https://inflammregen.biomedcentral.com/articles/10.1186/s41232-017-0045-6 https://www.hindawi.com/journals/sci/2021/6546780/ https://www.ncbi.nlm.nih.gov/pmc/articles/PMC7339207/  https://www.ecmh.org/journal/view.php?number=1579

: As you requested, we introduced the existing clinical trials using MSCs for liver disease and their results in the section 5 in the revised manuscript: “5. MSCs-based clinical application for liver disease.

Fifty-six clinical trials using MSCs for liver disease, such as cirrhosis, acute liver failure, and hepatitis, have been reported and the majority of these were registered in Asian countries [106]. MSCs that are mostly obtained from bone marrow, umbilical cord, and adipose tissue are used in clinical trials of the liver, and injected through peripheral vein or hepatic artery [107]. The number of cells and frequency of injection is various, and the MSC effect on liver fibrosis is inconsistent. Body-weight based dosing within the range (0.5 × 106 - 3 × 106 cells/kg) for a single dose is used in most clinical trials, where some studies use total MSC quantity (1 × 107 - 20 × 107 cells). Dose as low as 1 × 107 MSCs showed the significant attenuation of liver fibrosis in some cases [108], while higher dose of 2 × 108 MSCs rarely improved the liver fibrosis in others [109]. However, more than 2 × 108 MSCs reduced liver damage in patients with liver cirrhosis [110]. In most clinical studies, one-time doses was administrated, but had no significant difference in treating liver cirrhosis compared with two doses a month apart [111]. Nevertheless, most clinical trials have showed that MSC based therapies have beneficial effects on liver fibrosis (Table 2). In patients with alcoholic liver cirrhosis, hepatic arterial injection of BM-MSCs significantly improved liver function and reduced collagen accumulation. [28,112]. Patients with HBV-induced liver cirrhosis had the alleviated expression of fibrotic markers and remarkable decrease of model of end-stage liver disease (MELD) score at 48 weeks after MSC injection, although the level of HBV DNA in serum was not changed [113,114]. It was also reported that MSC treatment attenuated liver cirrhosis by exerting their immunomodulatory properties. MSC transplantation upregulated immunomodulatory factors, such as serum M-colony stimulating factor, macrophage migration inhibitory factor, and IL-18, and regulated imbalance of Treg/Th17 cell [108,115]. In addition, protective effect of hepatocyte-like cells differentiated from MSCs was reported in several clinical trials. Hepatocytes-lineage committed MSCs presented similar protective effects like undifferentiated MSCs [116], and they improved liver function in patients with liver cirrhosis or end-stage of liver disease [116-118]. Furthermore, no side effects and complications were observed in the above clinical studies except for mild fever that occurred after MSC injection. The mild fever disappeared within 12 hours after MSC treatment. However, two papers have reported that MSCs rarely have therapeutic effect in patients with liver cirrhosis; Mohamadnejad et al. [109] presented that MSC injection hardly reduced liver damage compared with the placebo group at 48 weeks after injection. Kantarcioglu et al. [119] also showed that scores of MELD and Child-Pugh in cirrhosis patients did not changed before and after MSC treatment. Therefore, it is required to conduct further study for large-scaled clinical trials encompassing multiple conditions, such as a wider range of doses and frequencies, and various administration routes, to establish the effective therapeutic dose and frequency for the clinically safe and long-term effect of MSC in liver disease.”

However, most clinical trials for liver fibrosis use MSCs, not MSC cell-free. MSC cell-free therapies are conducted in various diseases, such as dry eye symptom (NCT04213248), T1DM is currently underway (NCT02138331) and neurological disease (NCT04202783, NCT04202770, NCT04388982), except for liver disease. Hence, no clinical trial using MSC-secretomes was described briefly, “However, there are currently no clinical trials for liver disease.” in revised manuscript (section 6 in the revised one).

  1. The report may include information as to which source of MSCs or MSCs derived exosomes is best for the treatment of liver fibrosis/cirrhosis and why?

: MSCs are obtain from various organs/tissues, such as bone-marrow, adipose tissue, tonsil palatine, umbilical cord blood, and placenta etc, and these MSCs and MSCs-secretomes have showed the anti-fibrotic effect in the liver [Transl. Res. 2009, 154, 122-132/ J. Cell. Mol. Med. 2021, 25, 701-715/Mol. Ther. 2021, 29, 1471-1486/ IUBMB Life 2019, 71, 2020-2030/ Stem Cell Res. Ther. 2021, 12, 294]. However, it is hard to estimate and/or determine which one among MSC sources and MSC-secretomes is the best for liver fibrosis based on the animal or clinical results reported so far, because their therapeutic actions for liver fibrosis have not been proven under the same experimental conditions. To find the optimal MSC source and MSC-derived factors, the experimental conditions, such as experimental animal models of liver fibrosis, quality of MSCs and MSC-derived factors, and treatment methods and so on, should be the same. And then MSCs of different sources and MSCs-secretomes can be compared. It is such a huge experiment that no one seems to have done it. As we explained in the manuscript, MSC-mediated inhibition of HSC activation has been mainly proven in relatively constant in vitro systems. However, the anti-fibrotic effect of MSCs differs depending on the employed animal models and the experimental methods of MSC transplantation. Considering dynamic of MSC-EVs, it needs to set up the culture condition of MSCs to get the consistent quality of quantity of EVs. In addition, technologies for EV isolation are necessary to be determined. Therefore, in the manuscript, we proposed that the standardization of an experimental approach securing an optimal quality and quantity of MSCs and MSC-secretomes is necessary to ensure a consistent therapeutic effect of MSC and EVs.

Round 2

Reviewer 1 Report

Unfortunately the authors refused to address my concern properly. Adding a short sentence did not address my point. As I indicated this should be discussed in a seperate section. Otherwise I am not able to confirm the revision. 

Author Response

We are sorry for misunderstanding your comments. We asked the handling editor for advice, and described the effects of TGF-β and autophagy influx in therapeutic potential of MSCs in the revised manuscript, as his/her suggestion. TGF-β stimulation influences the biological and therapeutic potential of MSCs. TGF-β treatment was shown to increase MSC proliferation and their therapeutic efficacy for several diseases including type 1 diabetes, sepsis, and renal ischemia/reperfusion injury, not for liver fibrosis [Int Immunopharmacol. 2017 Mar;44:191-196. /Stem Cell Res Ther. 2020 Sep 3;11(1):378./Cytotherapy. 2019 May;21(5):535-545]. In addition, enhanced autophagy influx elevates therapeutic potential of MSCs in colitis damage, acute liver failure and immunosuppression. However, one article [Wang et al. J Cell Physiol. 2020 Mar; 235(3): 2722–2737] provide the contrary findings: upregulated autophagy influx alleviates therapeutic potential of MSCs in liver fibrosis. Based on these finding, we add the explanation for improvement of MSC therapeutic potential by TGF-β or autophagy in the revised manuscript, in line with the topic of the manuscript: “MSC engineering that modifies gene expression or metabolic process impacting biological functions of MSCs could increase proliferative and immunomodulatory properties of MSCs, enhancing their therapeutic function. Among factors engineering MSCs, TGF-β was shown to remarkably increase MSC proliferation [97-100]. TGF-β1 treatment elevated BM-MSCs proliferation by promoting nuclear localization of β-catenin in Smad3-dependent manner [99]. Kim et al. [100] reported that TGF-β1 stimulated expression of runt-related transcription factor 1, and extended self-renewal and proliferation of MSCs. MSCs engineered with TGF-β also reinforced their therapeutic efficacy for several diseases, such as type 1 diabetes, sepsis and renal ischemia/reperfusion injury, by increasing their immunomodulatory potential, although they rarely had effect on liver fibrosis. TGF-β-manipulated MSCs increased the insulin production and inhibited the expressions of pro-inflammatory cytokines in mice with type 1 diabetes [101]. Liu et al. [102] have revealed that MSCs overexpressing TGF-β have more favorable therapeutic effects by decreasing macrophage infiltration in cecal ligation and puncture-induced sepsis mice than MSCs. Administration of TGF-β-overexpressed MSCs restored renal function and attenuated inflammation against renal ischemia/reperfusion injury [103]. In addition, autophagy is known as a cell protection mechanism against various harmful damages, suggesting that autophagy influx promotes MSC viability, and contributes to the enhanced therapeutic potential of MSCs. Regmi et al. [104] demonstrate that elevated autophagy increases cell viability and decreases ROS generation of three-dimensional cultured MSCs (MSC3D), and the MSC3D alleviates dextran sulfate sodium-induced colitis damage more effectively than two-dimensional cultured MSCs (MSC2D) do. It was also shown that MSC3D had more resistance to severe oxidative stress than MSC2D did, and that MSC3D effectively modulated inflammation and improved therapeutic effect in the acute liver failure model [105]. In addition, autophagy regulates immunomodulatory effect of MSCs. Gao et al. [106] have revealed that rapamycin (autophagy inducer) increased immunosuppressive potential of MSCs, whereas 3-MA (autophagy inhibitor) reduced it. These findings suggest that enhanced autophagy influx in MSCs increase the therapeutic potential of MSCs. However, Wang et al. [107] presented that suppression of Beclin-1, essential mediator of autophagy, in MSC improved therapeutic and immunomodulatory properties of MSCs in CCl4-induced liver fibrosis model. The results are contrary to other findings. Therefore, further study is required to determine the effect of autophagy in MSC efficacy.”

Round 3

Reviewer 1 Report

I really appreciate the focus and accurate addressing the issue by the respected authors. I do not have any further comments